# SEMI-PARAMETRIC LANGUAGE MODEL WITH SELECTIVE MEMORY

## ABSTRACT

Pretrained on trillions of tokens, LLMs are known for their ability to store a large amount of factual knowledge in their parametric memory. However, recalling facts from this memory is known to be unreliable, particularly for long-tail knowledge—obscure facts infrequently mentioned in training data. Although retrieval-augmented generation (RAG) is the standard solution, it introduces overheads such as increased inference costs due to longer input contexts; preprocessing and indexing extensive document collections also adds additional engineering complexity. In this work, we propose a novel approach to improve the factuality of LLMs on long-tail knowledge. We begin by identifying *atomic facts*—short statements detailing relationships of some entities—that are not present in a pretrained LLM's parametric memory. These facts are then stored in an external, non-parametric memory. Subsequently, the model undergoes continuous pretraining, enabling it to learn *when* to consult this external memory at inference time. Compared with existing approaches, our method uses a compact external memory that selectively stores only the atomic facts not known to the LLM, resulting in minimal additional inference-time costs in terms of both time and space. Furthermore, our method outperforms fully trained models of comparable size on knowledge-intensive benchmarks by more than 10% on some benchmarks and achieves competitive results against larger models.

## 1 INTRODUCTION

Large language models (LLMs) have demonstrated an outstanding ability to learn a substantial amount of world language from their training corpus, storing this knowledge in their parameters and excelling in a wide range of applications. However, despite these advanced capabilities, LLMs frequently encounter the problem of hallucination, particularly when dealing with long-tail knowledge that is less represented in their training data (Mallen et al., 2023; Asai et al., 2023; Kandpal et al., 2023; Wei et al., 2024).

Recent work has explored integrating external memory into language models to improve factuality and reduce memorization at inference time (Li et al., 2025; Mallen et al., 2023; Li et al., 2024; He et al., 2023). However, a key limitation of this training-free approach is the need for a complex proposal procedure to decide between generating or retrieving from memory. Another line of work augments language model training with retrieval-based memory (Guu et al., 2020; Borgeaud et al., 2021). This type of methods force all retrieved knowledge into model parameters, which adds significant overhead during training and inference. Memory[3] (Yang et al., 2024) integrates externalized knowledge into attention layers by compressing key value representations. More recently, Large Memory Language Model (LMLM) (Zhao et al., 2025) introduced a pretraining recipe that stores factual knowledge both in model weights and in an external database. While effective, LMLM does not differentiate long-tail knowledge from sufficiently learned knowledge, which can lead to unnecessary retrievals and even degrade downstream performance.

A central challenge still remains: when and how to decouple long-tail knowledge during pre-training to improve factuality without sacrificing other capabilities. Offloading knowledge too early may limit the model's ability to form long-range associations between knowledge entities, while offloading too late may fail to sufficiently remove long-tail knowledge from the model parameters.

We address this with continual pretraining and selective memory, which uses partially trained models to detect knowledge gaps and adaptively externalize them. Our method begins by employing model-based or frequency-based methods to score long-tail knowledge within the training corpus. We preprocess the corpus into interleaved sequences of standard text and extracted knowledge segments, where the long-tail knowledge is stored in an external non-parametric memory. During continued pretraining, we implement an adaptive masking strategy: when the cross-entropy loss for a knowledge segment exceeds a predefined threshold, indicating potential memorization difficulty or long-tail knowledge, we masks these segments from the next-token prediction objective. To enable effective retrieval from this external memory during inference, we also finetune a lightweight query adapter that learns to generate appropriate query representations and maintains the separation between parametric knowledge (encoded in model weights) and non-parametric knowledge (stored in external memory).

Our method demonstrates superior performance on memory-intensive benchmarks compared to fully trained models of equivalent size, and even achieves competitive results compared to larger models. Compared to full offloading, this approach achieves a favorable trade-off: it preserves performance on general and reasoning-heavy benchmarks while shows clear improvements on long-tail knowledge tasks.

## 2 SEMI-PARAMETRIC LANGUAGE MODELS (SPLM)

### 2.1 IDENTIFY LONG-TAIL KNOWLEDGE

Common knowledge—frequent facts and patterns—are best captured in the model's parametric weights, allowing for fast, generalizable inference. Offloading these to external memory can force the model to rely on retrieval for even simple knowledge, increasing inference time and reducing generalizability. On the other hand, long-tail knowledge is difficult to internalize, requires repeated exposures, and offers little generalization benefit, yet it consumes disproportionate model capacity. Following Zhao et al. (2025), we pre-processed the training corpus into interleaved regular text and memory segments using a finetuned extraction model. Each memory segment contains a query–answer pair wrapped in special tokens: m_start [query text] m_retrieve [answer text] m_end , where m_start marks the beginning of a memory segment, and the m_retrieve token indicates the start of the answer which should be looked up from external memory. For example, the sentence "Sugeno received the IEEE Frank Rosenblatt Award in 2010 for his contributions to the field of fuzzy systems." is preprocessed as follows:

> Sugeno received the m_start Michio Sugeno Award Received m_retrieve IEEE Frank Rosenblatt Award m_end IEEE Frank Rosenblatt Award in m_start Michio Sugeno Year of Award m_retrieve 2010 m_end 2010 for his contributions to the field of fuzzy systems.

In addition, we score each memory segments in terms of how rare or hard the knowledge segments. We experiment with a few variants for proxy of long tail knowledge, including model based (loss of answer tokens) and model agnostic (frequency of entities). We leave the discussion on this part to Section 4.2.

### 2.2 TRAINING WITH SELECTIVE MEMORY

To prevent the model from inefficiently memorizing these rare or difficult knowledge pieces, we selectively delegate them to an external memory. In practice, this is achieved by masking out the answer tokens for hard knowledge segments during continued pre-training, so the model is trained to intermix free text generation with knowledge retrieval calls. This selective masking ensures that the model's parametric capacity is focused on common knowledge, while long-tail knowledge is handled via retrieval.

**Selective Memory** Offloading all knowledge to memory can bloat the retrieval store with trivial facts, slow down inference, and make the system brittle when retrieval fails. Prior work has shown that pretrained models' loss is a reliable proxy for knowledge gaps Feng et al. (2024), where persistent high loss usually indicates unreliable memorization. For a given memory segment $s_i$, let $A_i$ denote the set of token positions corresponding to its answer text. We compute

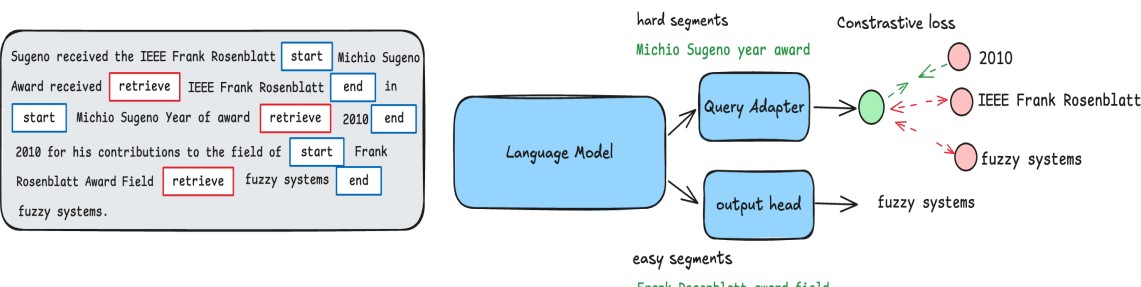

Figure 1: Training with selective memory and query adapter. Easy memory segments are trained directly with the standard next-token prediction loss, while hard segments are routed through a lightweight query adapter. The query adapter is optimized with a contrastive loss: embeddings for query span is pulled closer to the correct answer embedding (positive) and pushed away from incorrect answers (negatives) drawn from other segments within the same document.

the average negative log-likelihood of the answer tokens $\ell_i$ as a proxy for the difficulty of the memory segment. We define hard memory segments as those with average loss above a threshold $S_H = \{s_i : \ell_i > \tau\}$. If the model finds a particular memory segment consistently difficult (loss beyond $\tau$), we mark that segment as a candidate for externalization. In practice, we experiment with either a fixed threshold (only mask answer span when the loss is above a fixed threshold) or an adaptive threshold (mask answer span when the loss is in the top $p\%$ of all the memory segments seen so far in training). For a memory segment of length $T$, we introduce a binary mask $\mathbf{M} = [m_1, m_2, \ldots, m_T]$ where $m_t = 1$ if $s_i \in \{S_H\}$ and $x_t$ is between $\boxed{\text{m\_start}}$. We apply this mask to the language modeling objective, so that the model is trained only on unmasked tokens. $\mathcal{L}_{\text{masked}} = -\frac{1}{\sum_{t=1}^{T} m_t} \sum_{t=1}^{T} m_t \cdot \log P(x_t \mid x_{<t}; \theta)$. This effectively removes the loss contributions from the answer spans of hard memory segments, discouraging the model from attempting to memorize them and instead encouraging reliance on external memory for factual recall. This adaptive masking mechanism enables the model to allocate its parametric capacity to common knowledge and compositional reasoning, while selectively offloading long-tail or hard-to-learn knowledge to a non-parametric memory.

**Query adapter** While the above mechanism teaches the model when to retrieve, we also need to train how to retrieve the correct information from the external memory. To enable efficient and robust retrieval at inference time, we introduce a lightweight, contextualized MLP query adapter that projects both query context and answer text into a shared embedding space where vectors of relevant pairs of query and answer are close to each other. The adapter is trained to map the high-dimensional contextual representation of a query (the pooled embeddings over the query span) to a lower-dimensional dense query embedding that can be used to retrieve relevant answers. The final hidden representation at the position of the $\boxed{\text{m\_retrieve}}$ token is passed through the adapter to obtain a dense query embedding $z_{q_i}$. Similarly, for each memory segment's answer, we obtain a dense answer embedding by applying a pooling function over the LLM's final hidden states for the answer text to obtain $z_{a_i}$. We train the adapter using a contrastive InfoNCE loss: For a batch of $N$ memory segments (i.e., $N$ query–answer pairs $(q_i, a_i)$), we treat each $(q_i, a_i)$ as a positive pair and all other $N-1$ answers in the batch as negatives for $q_i$:

$$\mathcal{L}_{\text{InfoNCE}} = -\frac{1}{N} \sum_{i=1}^{N} \log \frac{\exp\left(\text{sim}(z_{q_i}, z_{a_i})/\tau\right)}{\sum_{j=1}^{N} \exp\left(\text{sim}(z_{q_i}, z_{a_j})/\tau\right)}, \tag{1}$$

where $\text{sim}(u, v) = \frac{u^\top v}{\|u\|\|v\|}$ denotes the cosine similarity between vectors and $\tau > 0$ is a temperature hyperparameter controlling the sharpness of the distribution. We leverage in-batch negatives during training, since each batch typically contains multiple memory segments extracted from the same document or entity, naturally serving as hard negative examples. The full training process is

described in Figure-1. Finally, the adapter loss is added on top of the training loss through:

$$\mathcal{L} = \mathcal{L}_{\text{masked}} + \alpha_{adapter}\mathcal{L}_{\text{InfoNCE}} \tag{2}$$

**Inference** At inference time, if the model outputs the special $\boxed{\text{m\_retrieve}}$ token, this signals that a memory retrieval is needed for the current query. At that point, we take the model's current hidden state over the query span and feed it through the adapter to get a contextualized query embedding $z_q$. This embedding is then used to perform a nearest-neighbor search in the external memory store for the most relevant stored answer.

## 3 EXPERIMENTS

### 3.1 EXPERIMENT SETUP

**Training details** We pretrained a baseline LLAMA-3-1B model for 150k steps on the DCLM corpus (220B tokens). We continued pretraining on the Wikipedia subset of the Dolmino corpus (3.7B tokens) for an additional 20k steps, using a learning rate of $4 \times 10^{-4}$, batch size of 4, and sequence length of 4,096. For the query adapter, we use a two-layer bottleneck MLP where both the hidden dim and output dim is set 384. The loss weight for the adapter ($\alpha_{adapter}$) is set to be 1.

**Inference details** For generation, we compare using an off-the-shelf sentence embedding model (`all-MiniLM-L6-v2` (Reimers & Gurevych, 2019; Wang et al., 2020)) to embed queries against using a trained query adapter. For the query adapter, we build the retrieval store for each memory segment in the corpus, where answer embedding generated by the adapter is used as retrieval key, and the answer tokens are used as retrieval value. For the sentence embedding model, we use a retrieval threshold of $0.6$, and for the query adapter we apply a higher threshold of $0.7$. To support efficient approximate nearest-neighbor search, we employ a FAISS IVF index (Johnson et al., 2019), which clusters vectors into 16,384 centroids, and we set `nprobe`$= 64$ during retrieval to balance recall and efficiency. For all benchmarks, we use answer level recall (ALR) as the evaluation metrics.

### 3.2 EVALUATIONS

We evaluate our models on a set of benchmarks designed to test general or long-tail factual knowledge in language models. For general-purpose QA, we adopt widely used benchmarks that serve as guardrails for assessing open-domain question answering: TRIVIAQA (TQA) (Joshi et al., 2017), NATURALQUESTION (NQ) (Kwiatkowski et al., 2019), ENTITYQ (Sciavolino et al., 2021) and HOTPOTQA (HQA) (Yang et al., 2018). These datasets emphasize broad coverage of knowledge and multi-hop reasoning, making them representative tests of general QA ability.

To specifically target long-tail factual knowledge, we use a suite of benchmarks including POPQA (Mallen et al., 2023), HEAD-TO-TAIL (Sun et al., 2024), and SIMPLEQA (Wei et al., 2024). POPQA is curated to cover questions on widely recognized entities and facts, while also incorporating a substantial proportion of less common, long-tail items. HEAD-TO-TAIL explicitly measures the popularity of entities: it consists of 18K question–answer pairs categorized into *head* (frequent), *torso* (moderately frequent), and *tail* (rare) entities. SIMPLEQA is a fact-seeking benchmark of short, unambiguous questions designed to challenge SOTA models.

### 3.3 BASELINES

**Vanilla training** As a baseline, we continued pretraining the model on the original Wikipedia subset of Dolmino for the same number of steps, without introducing any external memory or selective masking. This setup measures the effect of additional domain-adapted pretraining alone, isolating gains that come purely from further exposure to the corpus.

**Full Memory** Following the setup in the LMLM framework (Zhao et al., 2025), we construct a memory-only baseline in which all factual tokens are offloaded to external memory. In this setting, the model never learns to parametrize knowledge internally; instead, retrieval is performed at every

`m_retrieve` call using a separate embedding model and fuzzy string matching against the memory index.

## 3.4 Main Results

| Model | TQA | NQ | HotpotQA | EntityQ | PopQA | Head-to-Tail | | SimpleQA |
|---|---|---|---|---|---|---|---|---|
| | | | | | | All | Tail | |
| LLaMA3 8B | 62.92 | 32.05 | 26.16 | 31.16 | 22.65 | 14.77 | 10.74 | 4.76 |
| LLaMA3 1B | 40.94 | 18.48 | 21.01 | 14.12 | 14.30 | 7.78 | 6.22 | 2.27 |
| LLaMA3 1B - 150k | 31.72 | 17.37 | 18.10 | 17.34 | 14.54 | 7.51 | 5.30 | 2.87 |
| +Vanilla training | 23.72 | 9.39 | 13.45 | 12.43 | 11.13 | 6.26 | 4.49 | 2.36 |
| +Memory | 24.64 | 14.71 | 13.06 | 23.95 | 23.04 | 10.29 | 10.09 | 8.58 |
| +SM (75%) | 26.79 | 16.04 | 15.72 | 25.84 | 21.81 | 11.37 | 10.58 | 8.97 |
| +SPLM | **28.65** | **16.24** | **16.86** | **26.38** | **25.37** | 11.3 | 10.28 | **11.35** |

Table 1: Answer-level recall (ALR) on open-domain QA benchmarks. The baselines include LLaMA3 8B, LLaMA3.2 1B, and a partially trained LLaMA3 1B (150k steps). For selective memory (SM), we report the result with an adaptive threshold of 0.75 on quantile of average answer token losses (25% of hard segments are included in memory).

In Table 1 we show the results on the answer-level recall (ALR) of a LLaMA-3 1B trained under different paradigms: continual pretraining on regular corpus (vanilla training), continual pretraining with fully offloaded memory (Memory), continual pretraining with selective memory (SM), and our full approach with selective memory combined with a query adapter (SPLM).

**Continual pretraining with memory outperforms fully trained models on knowledge intensive tasks** Compared to a fully trained Llama 1B model, training with memory demonstrates superior performance particularly on benchmarks that emphasize long-tail or entity-centric knowledge, such as EntityQ (17.34% for fully trained model vs), PopQA (14.54% for fully trained model vs 23.04% with memory), and head-to-tail. It also achieves comparable results compared to a much bigger model (Llama 8B), indicating that externalizing long-tail knowledge can enhance factual recall without increasing model size.

**Selective memory outperforms full knowledge offloading** We also observe clearly improvements when using selective memory compared to fully offloading memory. While offloading all memory improves recall on certain long-tail benchmarks such as PopQA, Head-to-tail and SimpleQA, it leads to degradation on general QA datasets (TQA, NQ) or those require multi-hop reasoning (HotpotQA). In contrast, selective memory achieves a better balance: it maintains competitive performance on general QA datasets while further improves the performance on long-tail benchmarks. This shows that adaptively masking only the hardest segments allows the model to preserve its parametric capacity for common knowledge and reasoning, while only relying on external memory for rare knowledge. Finally, incorporating a lightweight query adapter on top of selective memory (SPLM) recovers much of the degradation observed on general QA benchmarks while further improving long-tail factual recall. We observe the degradation on certain benchmarks is a result of the model generating imperfect and unnecessary retrieval as shown in the sample generations in Appendix C. Selective memory combined with query adapter mitigates this issue by learning to produce contextualized, discriminative, and richer query embeddings that are naturally aligned with the model's internal representation.

## 4 Ablations

### 4.1 When should we decouple factual knowledge?

**LMs are inefficient in memorizing factual knowledge in parametric memory** In Figure 2 we plot the answer-level recall (ALR) of a Llama3 1B and Llama 8B model trained on DCLM corpus as training progresses. We observe that some benchmarks exhibit clear improvements with additional training, while others remain stagnant. For example, NQ and HotpotQA continue to benefit from

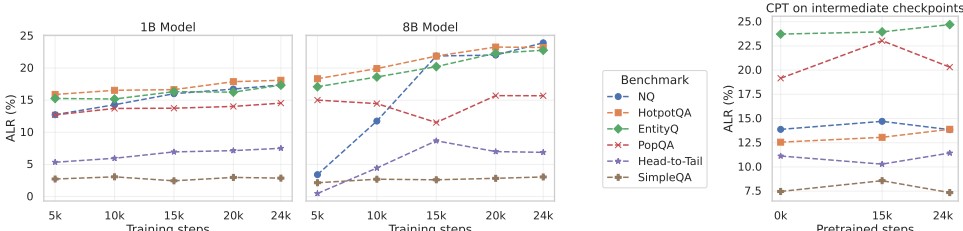

Figure 2: Left: Answer-level recall (ALR) across training steps when the model is trained on regular corpus. Right: Continual pretraining (CPT) on memory corpus starting from an pretrained model at intermediate checkpoints.

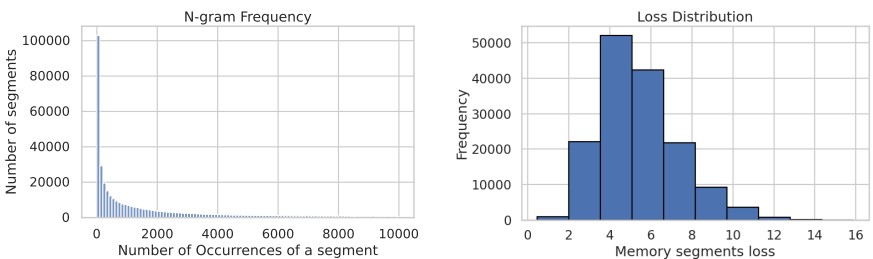

Figure 3: Model based and model agnostic scoring for long-tail knowledge.

longer training, showing steady gains up to 240k steps. By contrast, simpleQA and popQA plateau quickly, indicating that continued training on the same corpus does little to improve performance on factual benchmarks. This is consistent with previous observation Chang et al. (2024) that factual knowledge are prone to forgetting and pretraining on more data shows no significant improvement.

**Pretraining vs. Continual Pretraining** Decoupling long-tail facts to external memory can be done during pre-training or during continuous pre-training time. However, it's hard to separate "factual memory" from "reasoning" in a clean way. Many reasoning capabilities are scaffolded and interleaved with factual text. For example, solving multi-step questions in math often relies on domain-specific facts or theorems embedded in the training data. If large spans of such text are masked, the model may be under-trained on certain compositional patterns, hurting downstream reasoning. In figure 2, we plot the performance of the final model when starting from an intermediate model trained on regular corpus for x steps and then continual pretrain on memory corpus for 20k steps. We observed that the final performances when training from an intermediate checkpoints are better than training from memory corpus from scratch.

## 4.2 DIFFERENT PROXIES FOR LONG-TAIL KNOWLEDGE

In this section, we compare model-based proxies and model-agnostic proxies for detecting long-tail knowledge in a corpus. Model-agnostic approaches estimate rarity directly from corpus statistics. One effective tool is *Infinigram* (Liu et al., 2024), which computes exact $n$-gram frequency distributions in a given corpus. We count the frequency if either the objective or the subjective entity in the memory segment appear in the corpus. While Infinigram provides an exact measure of $n$-gram frequencies, its distribution is extremely skewed (Figure 3), making it challenging to draw a clear boundary between head and long-tail knowledge. Alternatively, we also experimented with model based scoring, where we prompt the Llama3-70B model (see Appendix for prompt details) to assign a popularity score to each knowledge segment on a scale from 1 to 10, with higher scores corresponding to more frequently occurring knowledge and lower scores marking long-tail or rare facts. We observe the same trend: the model assesses most of the knowledge segments as long-tail knowledge.

On the other hand, model based proxies such as training loss provides a natural signal of difficulty and has been widely used for curriculum learning. Figure 3 shows the distribution of memory segment losses at the start of continual pretraining: most losses concentrate around moderate values, while the long tail represents knowledge the model struggle to internalize.

As shown in 3, knowledge frequency tend to display Zipf law, where most memory segments are rare and low frequency. In contrast, the distribution under loss-based scoring is closer to normal, with many segments falling into a moderate difficulty range rather than being pushed to the extremes. This discrepancy suggests that, despite the frequency of appearing in the corpus, the model exhibits heterogeneous difficulty in internalizing difficult factual knowledge. This motivates us to use model-based proxies at continual pretraining time to distinguish hard knowledge that the model struggles with.

## 4.3 DIFFERENT MEMORY THRESHOLD

**Fixed vs adaptive threshold**  In the simplest variant, we apply a fixed threshold $\tau$, masking any answer segment with average loss above $\tau$. This provides a straightforward knob on how aggressively long-tail knowledge is offloaded: the higher $\tau$ is, the fewer segments are delegated to external memory. Alternatively, we experiment with an adaptive threshold, where at each step we compute the distribution of average losses across all answer segments seen so far and mask those falling in the top $\tau\%$ (e.g., 75th percentile). The adaptive strategy dynamically adjusts to training progress and corpus difficulty: early in training, more segments are masked, while later only the most challenging segments are delegated to memory. We report answer-level recall (ALR) under different fixed thresholds shown in Figure 4. For fixed threshold, as the memory threshold increases, ALR generally decreases for the entity-centric and longtail benchmarks such as EntityQ and PopQA, but increases for the common knowledge QA datsets such as NQ and HotpotQA. This aligns with our assumption that aggressive memory offloading (lower threshold) could hurt performances on general benchmarks. On general QA benchmarks such as TQA, NQ, and HotpotQA, we observe a positive trend as the adaptive memory threshold increases, whereas performance on long-tail benchmarks is comparatively stable. This pattern aligns with our hypothesis: by delegating only the hardest factual segments to external memory, the model preserves parametric capacity for common knowledge and improves generalizability, while maintaining strong recall on rare, entity-centric queries.

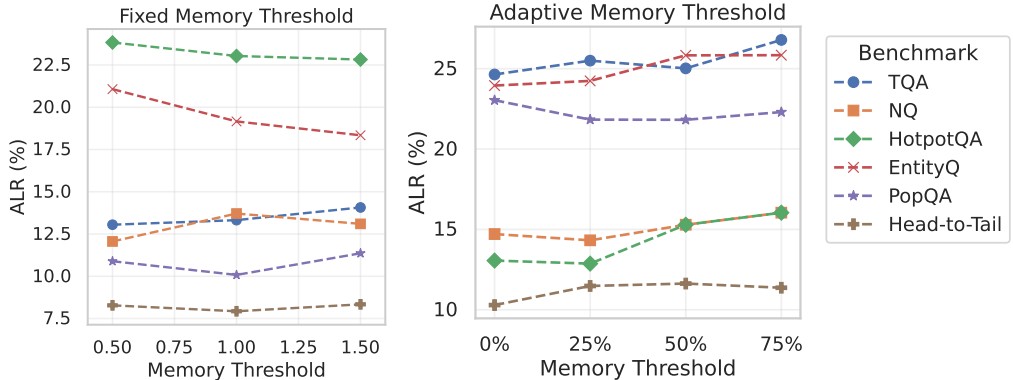

Figure 4: Answer level recall (ALR) for different memory threshold. The higher the threshold is, the fewer segments are delegated to external memory. Left: Fixed memory threshold. Right: Adaptive memory threshold.

**Trade-off between accuracy and latency**  Long-tail datasets such as POPQA and SIMPLEQA trigger retrieval far more frequently than other benchmarks, indicating that the model is more reliant on external memory for long tail questions. While one might expect retrieval frequency to increase linearly with more higher thresholds, this is not observed in practice. The reason is that segment losses are highly skewed and clustered as shown in previous histogram, so adjusting thresholds often has non-linear effects: small changes may have little impact or may trigger sudden jumps in retrieval.

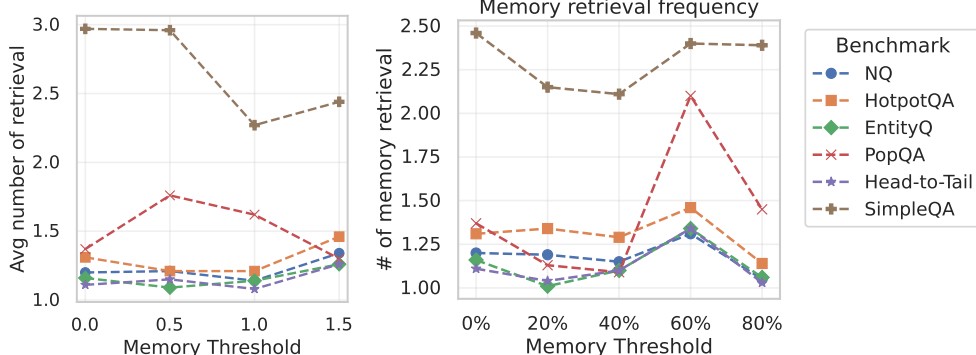

Figure 5: Average number of retrieval calls for different memory threshold, left is fixed threshold, and right is adaptive threshold. The higher the threshold is, the fewer segments are delegated to external memory.

## 5 RELATED WORK

**Non-parametric language modeling**  One recent line of research has explored integrating non-parametric memory with language models generation to reduce memorization and improve factuality. The earliest work includes kNN-LM, which interpolates language model generation and retrieved nearest neighbors at inference time to adjust output distribution (Khandelwal et al., 2019). More recent inference-time only approaches include Chunk-Distilled LM (Li et al., 2025), which speculates multi-token chunks using a retrieval datastore to accelerate generation, as well as REST (He et al., 2023) and NEST (Li et al., 2024), which combine speculative decoding with retrieval to improve efficiency and factuality.

**Retrieval-augmented pretraining**  Several recent approaches have explored incorporating retrieval directly into the pretraining process. RETRO (Borgeaud et al., 2021) demonstrated that integrating retrieved context during pretraining improves generalization and reduces undesired memorization by allowing smaller models to match the performance of much larger purely parametric LMs. More recently and most relevant to us, Zhao et al. (Zhao et al., 2025) introduced Large Memory Language Models (LMLM), which explicitly separates factual knowledge storage from model weights by offloading specific factual details to an external database during pre-training. Our approach shares a similar motivation, but instead of we leverage signals from a partially trained model to adaptively decide what knowledge to offload.

## 6 CONCLUSION

We introduced semi-parametric language models with selective memory (SPLM), a framework that improves factuality on long-tail knowledge without sacrificing general capabilities. By masking high-loss segments and delegating them to external memory, SPLM preserves parametric capacity for common knowledge and reasoning while offloading rare facts. A lightweight query adapter enables contextualized retrieval aligned with the model's representations. Experiments demonstrate that SPLM achieves a favorable trade-off, outperforming memory-only baselines and rivaling larger models. This highlights selective memory as a promising direction for scaling factuality in LLMs efficiently.

There are several interesting directions for future research. We primarily targeted factual recall on short QA benchmarks, but the framework could naturally extend to other domains such as mathematical theorems or code snippets that occur in training data. An interesting question for future work is whether decoupling knowledge in this way could also improve performance on reasoning-intensive benchmarks. By externalizing rare factual content, the model could dedicate more of its parametric capacity and compute to learning reasoning strategies rather than memorizing infrequent facts. Another interesting direction would be to explore joint training of the adapter and the language

model, which may further improve performance by enabling integration of external knowledge during inference.

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

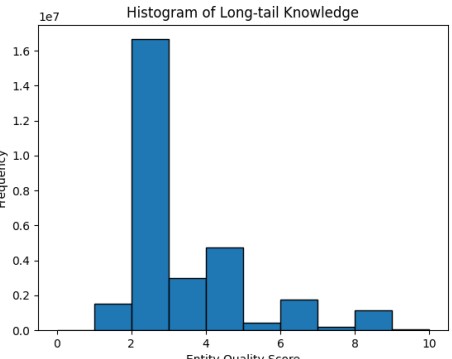

Figure 6: Histogram of long tail scoring using Llama3-70B. 1 indicates low popularity and 10 indicates high popularity.

## A   APPENDIX

## A   ADDITIONAL IMPLEMENTATION DETAILS

## B   MODEL BASED SCORING

### B.1   PROMPT USED FOR LLAMA3 70B

> You are a capable model that can determine, based on the atomic factual knowledge query, whether the atomic factual knowledge belongs to a long-tailed (niche or less commonly referenced) domain. To assess the entities popularity, use the following rules: 1) The number of times the atomic factual knowledge potentially can referenced or cited by other documents on internet. 2) The number of times the atomic factual knowledge potentially can be linked to or duplicated on internet. Please rate the atomic factual knowledge with a score between 1 and 10, where 1 indicates low popularity (long-tailed knowledge) and 10 indicates high popularity (widely referenced knowledge). Return the score in JSON format, for example: {"quality_score": predicted score } text:

Interestingly, as shown in Figure 6, the model rates most extracted factual knowledge as long-tail, which aligns closely with the skewed $n$-gram frequency distribution estimated by Infinigram (Figure 3). In contrast, the distribution under loss-based scoring is closer to normal, with many segments falling into a moderate difficulty range rather than being pushed to the extremes. This discrepancy suggests that, despite the frequency of appearing in the corpus, the model exhibits heterogeneous difficulty in internalizing difficult factual knowledge.

## C   SAMPLE GENERATIONS FROM SIMPLEQA

**Q1.** When was the exact day, month, and year the fpt university was founded?

**Gold Answer:** september 8, 2006

**Full Memory:** The university was founded in m_start University of California, Berkeley Founded In m_retrieve 1868 m_end 1868.

**Selective memory:** m_start FPT University Founded On m_retrieve september 8, 2006 m_end september 8, 2006.

**Q2.** Q: What was the word of the decade (2010–2019) according to the american dialect society?

**Gold Answer:** They.

**Full Memory:** ⬚m_start⬚ Q: What was the word of the decade (2010–2019) Answer ⬚m_retrieve⬚ web ⬚m_end⬚ web

**Selective memory:** The word of the decade was ⬚m_start⬚ Word of the Decade 2010-2019 ⬚m_retrieve⬚ they ⬚m_end⬚ they, according to the american dialect society.

**Q3.** Q: What was the first name of the irish-born computer programmer kathleen rita mcnulty mauchly antonelli's father?

**Gold Answer:** james mccullough

**Selective memory:** He was called ⬚m_start⬚ Kathleen Rita McNulty FatherName ⬚m_retrieve⬚ troy and claudette mcghee ⬚m_end⬚ troy and claudette mcghee.

**SPLM (ours):** She was called ⬚m_start⬚ Kathleen Rita Mcculty Mauchly FatherName ⬚m_retrieve⬚ james mccullough ⬚m_end⬚ james mccullough

**Q4.** Q: Who was the 9th chief minister of jammu and kashmir?

**Gold Answer:** mehbooba mufti

**Selective memory:** ⬚m_start⬚ 9th Chief Minister of Jammu and Kashmir Name ⬚m_retrieve⬚ mehbooba mufti ⬚m_end⬚ mehbooba mufti

**SPLM (ours):** Bakhsh Singh, who was the 9th chief minister of jammu and kashmir from ⬚m_start⬚ Bakhsh Singh Term Start Date ⬚m_retrieve⬚ 9 april 1964 ⬚m_end⬚ 9 april 1964 to ⬚m_start⬚ Bakhsh Singh Term End Date ⬚m_retrieve⬚ 10 july 1970 ⬚m_end⬚ 10 july 1970.

