# OpenReview forum: "Semi-parametric language model with selective memory"
_ICLR.cc/2026/Conference — Submitted to ICLR 2026_

### Official Review · Reviewer_1XW6 · 2025-10-21

**Soundness:** 1
**Presentation:** 2
**Contribution:** 2
**Rating:** 2
**Confidence:** 4

**Summary:**

The paper proposes a semi-parametric language model (SPLM) that keeps common knowledge in model weights and offloads long-tail facts to a compact external memory. During continued pretraining, the model masks tokens for hard factual spans and learns to emit a special token to trigger retrieval. A lightweight query adapter is trained with a contrastive loss so the hidden state at the retrieval point can fetch the right memory entry. Experiments on open-domain QA benchmarks show modest gains on tail-heavy datasets compared to a similar-size baseline, and some ablations vary the masking threshold. However, the conceptual novelty over prior retrieval-augmented pretraining and LMLM-style designs seems limited, and several crucial details are underspecified.

**Strengths:**

1. Clear motivation to avoid storing rare facts in parameters and to reduce unnecessary retrieval for common facts.
2. Simple training recipe (selective masking + small adapter) that is easy to implement on top of standard continued pretraining.
3. Compact external memory and selective retrieval reduce the runtime overhead compared to full RAG contexts.

**Weaknesses:**

1. The approach is very close to existing retrieval-augmented pretraining and large-memory LM work; the paper does not articulate a distinctive technical contribution beyond selective masking.
2. Evaluation is narrow (mostly short-form QA). There is little analysis of failure cases, retrieval precision/recall, or robustness to distractor memories.
3. This method has not been shown to scale to larger LLM backbones, nor has it been demonstrated to be necessary for them. In general, larger language models tend to have fewer long-tail tokens, so addressing these long-tail distribution issues may bring little improvement for QA tasks.

**Questions:**

1. Could the authors please explain how the memory used for retrieval is constructed? In other words, what is the complete set of negative samples used in the contrastive loss when training the adapter?
2. Could the authors provide experimental evidence or theoretical justification showing that the proposed method can scale to larger models? Additionally, does this method affect the model’s normal instruction-following ability or its general reasoning capability?

---

### Official Review · Reviewer_uoqn · 2025-11-01

**Soundness:** 2
**Presentation:** 2
**Contribution:** 2
**Rating:** 4
**Confidence:** 4

**Summary:**

This paper proposes “Semi-Parametric Language Models with Selective Memory” (SPLM), a method intended to improve long-tail factual recall without substantially increasing inference overhead or sacrificing general language understanding. The authors start from a partially trained LLaMA-based model and identify high-loss or low-frequency knowledge segments, which are masked out and stored in a compact external memory. Continued pretraining is then performed using an adaptive masking strategy that trains the model to rely on its internal weights for common knowledge while learning to query the external store for rare facts. A lightweight query adapter maps the model’s hidden state at special retrieval tokens to dense embeddings for nearest-neighbor search in the external memory. Experiments on several QA benchmarks suggest that the approach improves answer recall on long-tail datasets (POPQA, SIMPLEQA, tail subsets of Head-to-Tail) while maintaining or modestly improving performance on general benchmarks (Natural Questions, HotpotQA) compared to a standard LLaMA baseline, a memory-only offloading model, and an 8B-parameter LLaMA. The paper includes ablation studies on the choice of thresholds for masking and compares model-based versus corpus frequency proxies for detecting long-tail facts.

**Strengths:**

* Focused long-tail handling: Proposes masking high-loss or low-frequency facts during continued pre-training, offloading them to a compact external memory. This improves recall on rare facts while preserving general reasoning.
 * Efficient retrieval via lightweight adapter: The query adapter trained with contrastive loss produces contextual embeddings for retrieving the correct fact efficiently, helping to avoid unnecessary or erroneous lookups.

**Weaknesses:**

* Incremental novelty: While the selective offloading of hard examples is effective, it builds closely on prior work. Selective memorization methods in continual learning (SeMem[A]) already use model loss to decide which samples to store in a non‐parametric memory, demonstrating scalability gains. Similarly, LMLM (Zhao et al., 2025) masks retrieved factual values during pretraining to enable lookup rather than memorization, and hierarchical memory approaches explicitly separate common vs. long‐tail knowledge. The core idea of masking hard knowledge segments to offload them is conceptually in line with these techniques. The paper needs a clearer positioning that distinguishes SPLM’s contribution from these existing approaches.

* The paper proposes both frequency‐based and model‐based (loss or LLM popularity scoring) proxies for long‐tail knowledge. However, the use of cross‐entropy loss as an indicator of memorization difficulty echoes previous curriculum‐learning and selective memory strategies. It would strengthen the contribution to show that the adaptive combination of proxies (e.g., loss quantiles plus LLM‐prompted popularity scores) offers a measurable advantage over purely frequency‐based or purely loss‐based scoring.

* Retrieval‐augmentation frameworks often fail to differentiate between queries needing long‐tail knowledge and those covering common knowledge. SPLM implicitly addresses this via the m_retrieve token, but the authors could expand on how the approach compares with systems that dynamically measure “long‐tailness” at inference time (e.g., by GECE metric [B]).

 * It is worth clarifying how the masking strategy affects learning of complex reasoning patterns that incorporate rare facts. The paper notes that masking too much text could impair reasoning, particularly on multi‐hop tasks. Discussing how SPLM might interact with other forms of reasoning benchmarks or tasks beyond factoid QA would situate the approach within a broader context of semi‐parametric models.

 * The authors need to have an ablation on their architectural choices and training strategies: For example: contrastive loss and query adapter



* Important related works:

  [A] Semiparametric Language Models Are Scalable Continual Learners (Peng et al., 2023)

  [B] On the Role of Long-tail Knowledge in Retrieval Augmented Large Language Models (Li et al., 2024)

  [C] Pretraining with hierarchical memories: separating long-tail and common knowledge (Pouransari et al., 2025)

  [D] Adaptive Semiparametric Language Models (Yogatama et al., 2021)

  [E] Knowledge-in-Context: Towards Knowledgeable Semi-Parametric Language Models (Pan et al., 2023)

  [F] RET-LLM: Towards a General Read-Write Memory for Large Language Models, Modarressi et. al. 2024

  [G] MemLLM: Finetuning LLMs to Use An Explicit Read-Write Memory, 2024

**Questions:**

* Minor:
  * L39: Recent work has -> Recent works have
  * L61: we masks -> we mask
  * L151-2: The adapter thus  ... : the sentence needs to be corrected.

 * The submission looks to be in an incomplete state. Why APPENDIX has empty sections with titles?

---

### Official Review · Reviewer_89gj · 2025-11-03

**Soundness:** 3
**Presentation:** 3
**Contribution:** 3
**Rating:** 4
**Confidence:** 4

**Summary:**

The paper proposes a hybrid method for continued pre-training of LLM where the model "offloads" additional 'atomic' facts into an external database. This database is then used at inference time to look up long tail facts.

Concretely the authors show that on a 1B LLMs their results vs. clear baselines performs better. The baselines are strong and represent meaningful results, i.e. 1. "vanilla" continued pre-training, 2. Full Memory Networks, 3. pre-trained embedder. 4. their full method.

**Strengths:**

Main strengths:
1. An interesting approach to dynamically (and automatically) combine retrieval and inference in LLMs.
2. Strong results that balance long-range reasoning tasks as well as general QA.
3. Good discussion around ablations and methods used.

**Weaknesses:**

Main weaknesses:
1. Paper is incomplete. Appendix is missing. And Acknowledgement are the default text. The paper is clearly not finished.
2. One of the key elements of the paper are completely missing: How does one create atomic facts for the training dataset. That is probably the key question that are not discussed in the paper at all.
3. The model that is trained is limited to 1B model in the Llama family. Whereas nowadays other open source alternatives are much better suited: incl. Qwen (which has probably much better out of the box performance that would be interesting to compare to see if the result still holds).

**Questions:**

1. How do you create the atomic facts? That seems to be the hardest part of the setting?
2. What are the exact reproducibility settings for your experiments?
3. What are results on other, better more novel models?

---

> ### Author Response · Authors · 2025-12-03
>
> We thank the reviewer for their thoughtful reading and raising these important concerns.
> - $\textbf{missing appendix material}$. We acknowledge that the initial submission had presentation issues, where parts of the appendix were missing. We would like to clarify that the main paper already contains the essential implementation details, including training configuration, retrieval setup, and evaluation protocol. In the revised draft, we have fixed the acknowledgment section and added the missing appendix content, including additional qualitative sample generations that better illustrate model behavior, typical success and failure modes of the baseline approaches.
> - $\textbf{Extraction of atomic facts}$. We agree that the construction of atomic facts is a critical component and should have been described more explicitly. The extraction of atomic facts were built on the previous LMLM work. Specifically, atomic facts are extracted using a fined tuned model that converts raw text into short, structured (query, answer) pairs. While we do not introduce a new extraction method, our contribution lies in how these extracted facts are selectively masked and offloaded based on model difficulty signals during continual pretraining.
> - $\textbf{Generalization on model families}$. We understand the concern that the main empirical results are reported only on Llama3-1B model. This choice was primarily driven by practical computational constraints. In Figure 2 in the paper we showed that additional pre-training yields diminishing returns on benchmarks focusing on factual knowledge even for bigger models (8B model). Our method also relies on relative difficulty as a proxy for long-tail knowledge rather than absolute frequency, making it inherently scale-adaptive as model capacity increases.

---

### Official Review · Reviewer_rN9y · 2025-11-04

**Soundness:** 2
**Presentation:** 1
**Contribution:** 2
**Rating:** 2
**Confidence:** 4

**Summary:**

This paper introduces a semi-parametric language model framework called SPLM to improve factual recall of long-tail knowledge. The core idea is to selectively offload only the facts that a model struggles to memorize, which are identified using high loss signals during training. These facts are stored in an external non-parametric memory. The model is then continually pretrained using an adaptive masking technique that teaches it when to retrieve from this memory rather than generating text. A separate lightweight query adapter is finetuned to manage the retrieval process efficiently. The authors demonstrate that their method improves performance on several long-tail knowledge benchmarks against comparable baselines.

**Strengths:**

1. The paper's core concept of selective offloading based on model difficulty is intuitive. This presents a sensible alternative to standard RAG which retrieves indiscriminately or full memory models which offload everything.
2. The two-stage training process, first teaching the model when to retrieve via masking and then how via the adapter, is a clean way to decouple the tasks. Using a frozen LLM backbone for adapter tuning is efficient.
3. The experimental results convincingly show that SPLM outperforms the full memory baseline, especially in balancing general QA performance with long-tail recall. The improvements on benchmarks like PopQA and SimpleQA are notable.

**Weaknesses:**

1. The empirical evaluation is limited. All experiments are conducted on a 1B model. It is unclear if these findings generalize to larger, more capable models where the parametric memory is much larger and the definition of "long-tail" might change.
2. The paper criticizes standard RAG but surprisingly fails to include any RAG baseline in Table 1. Without this comparison, it is impossible to assess if the proposed complexity of selective memory and adapter finetuning is superior to simply applying a standard RAG method to the LLaMA3 1B model.
3. The paper appears unfinished and lacks crucial details. The appendix is mentioned, but key sections like "Additional Implementation Details" (Appendix A) and "Sample Generations" (Appendix B)  are entirely missing from the document. Only a prompt (Appendix C) is included.
4. This lack of detail is problematic. For example, the caption for Table 1 states an adaptive threshold of 0.6 was used , but the justification for this choice over others (explored in Section 3.5.3 ) is absent without the implementation details.
Minor:
1. Figure 1 on page 3 is too wide and clearly extends beyond the right page margin .

**Questions:**

See Weakness.

---

> ### Author Response · Authors · 2025-12-03
>
> We thank the reviewers for their thoughtful reading and detailed feedback.
> - $\textbf{Model scale and generalization}$. We understand the concern that the main empirical results are reported on a 1B model. This choice was primarily driven by practical computational constraints. In Figure 2 in the paper we showed that additional pre-training yields diminishing returns on benchmarks focusing on factual knowledge even for bigger models (8B model). Our method also relies on relative difficulty as a proxy for long-tail knowledge rather than absolute frequency, making it inherently scale-adaptive as model capacity increases.
> - $\textbf{Missing details in appendix material}$. We thank the reviewer for pointing out the missing subsections in the appendix, and we have included qualitative sample generations in the updated draft. We would like to clarify that the main paper already contains the essential implementation details, including training configuration, retrieval setup, and evaluation protocol.
> - $\textbf{Ablation on adaptive threshold}$. In response to the reviewers’ concerns, we have also included ablation studies on the adaptive memory threshold. We report results using an adaptive threshold of 0.6 in the main table as a representative point that balances factual recall and external memory size, but we emphasize that this is a hyperparameter that should be chosen based on the desired trade-off.

---

### Meta-Review · Area_Chair_dn1w · 2025-12-30

**Summary:**

This paper proposes to augment an LLM with an external explicit memory to contain all the fact that the model struggles to correctly learn and thus ``store'' in their parameters. This hybrid approach, called semi-parametric LLM (*), is evaluated on a number of QA benchmarks and compared against baselines using external memories and not.

The reviewers thought that, in principle, the overall idea of semi-parametric LLMs is interesting, but don't recommend the paper's publication in its current state because: a) they find the presentation poor and the paper rushed (rN9y, e.g., with missing appendix, 89gj), lack of baselines such as RAG and more recent works including memories (rN9y, uoqn, 1XW6) c) focus on smaller-scale models and only from a same family (rN9y, 89gj, 1XW6). I agree with all the points above, with the exception of (c) when it comes to ``small models only''. I hear the complaint from the authors' rebuttal on the computational cost of running larger LLMs. However, if the experimentation setting does not change, the claims in the paper need to be revised. To improve the setting, still on a computational budget, the authors can include small models from other families.

(*) I suggest authors to change the name, as "semi-parametric" has a well-defined meaning in statistics and probabilistic machine learning, which differs a lot from the one given in this paper.

**Reviewer Concerns:**

During the rebuttal the authors provided only two short messages to reviewers rN9y and 89gj.
They improved the presentation a bit, and added some appendixes (one, Appendix A appears to be still missing). And they added a small ablation study. However, all the other concerns regarding novelty and missing baselines remain unanswered.

**Reviewer Scores:**

Since there was only little (and a partial) discussion, I believe reviewers would have not increased their score significantly.
Reviewer rN9y might have increased the score from 2 to 4, but this would not change much the status of the paper.

---

### Decision · Program_Chairs · 2026-01-26

Reject